# Role of Natural Compounds Modulating Heme Catabolic Pathway in Gut, Liver, Cardiovascular, and Brain Diseases

**DOI:** 10.3390/biom14010063

**Published:** 2024-01-02

**Authors:** Sri Jayanti, Libor Vitek, Camilla Dalla Verde, John Paul Llido, Caecilia Sukowati, Claudio Tiribelli, Silvia Gazzin

**Affiliations:** 1Liver brain Unit “Rita Moretti”, Fondazione Italiana Fegato-Onlus, Bldg. Q, AREA Science Park, ss14, Km 163,5, Basovizza, 34149 Trieste, Italy or srij001@brin.go.id (S.J.); camilla.dallaverde@fegato.it (C.D.V.); johnpaul.llido@fegato.it (J.P.L.); or caecilia.sukowati@brin.go.id (C.S.); ctliver@fegato.it (C.T.); 2Eijkman Research Centre for Molecular Biology, Research Organization for Health, National Research and Innovation Agency, Cibinong 16915, Indonesia; 3Institute of Medical Biochemistry and Laboratory Diagnostics, and 4th Department of Internal Medicine, General University Hospital and 1st Faculty of Medicine, Charles University, 12000 Prague, Czech Republic; vitek@cesnet.cz; 4Department of Life Sciences, University of Trieste, 34139 Trieste, Italy; 5Department of Science and Technology, Philippine Council for Health Research and Development, Bicutan, Taguig City 1631, Philippines

**Keywords:** bilirubin, herbal medicine, NRF2, heme-oxygenase, MAFLD, neurodegeneration, nutraceuticals, Alzheimer’s disease, cancer, Parkinson’s disease

## Abstract

The crucial physiological process of heme breakdown yields biliverdin (BV) and bilirubin (BR) as byproducts. BV, BR, and the enzymes involved in their production (the “yellow players—YP”) are increasingly documented as endogenous modulators of human health. Mildly elevated serum bilirubin concentration has been correlated with a reduced risk of multiple chronic pro-oxidant and pro-inflammatory diseases, especially in the elderly. BR and BV per se have been demonstrated to protect against neurodegenerative diseases, in which heme oxygenase (HMOX), the main enzyme in the production of pigments, is almost always altered. HMOX upregulation has been interpreted as a tentative defense against the ongoing pathologic mechanisms. With the demonstration that multiple cells possess YP, their propensity to be modulated, and their broad spectrum of activity on multiple signaling pathways, the YP have assumed the role of an adjustable system that can promote health in adults. Based on that, there is an ongoing effort to induce their activity as a therapeutic option, and natural compounds are an attractive alternative to the goal, possibly requiring only minimal changes in the life style. We review the most recent evidence of the potential of natural compounds in targeting the YP in the context of the most common pathologic condition of adult and elderly life.

## 1. Introduction

Far from being only a diagnostic marker for liver diseases, mild elevation of the total serum bilirubin (TSB) concentration has been repeatedly correlated with a lower risk of developing chronic diseases typical of adult and elderly life [1,2,3,4,5,6]. The protection is mediated by bilirubin modulatory activity on the cellular redox state, immunity, cellular metabolism, body glucose, insulin balance, etc. Thus, bilirubin is now considered a biomarker of disease resistance, a predictor of all-cause mortality, and a molecule that can promote health in adults. A complex network links the enzymes involved in bilirubin production and multiple signaling pathways that may be targets of pharmacologic induction [2,4,5]. Based on that, there is an ongoing effort to induce their activity as a therapeutic option. Natural compounds are attractive alternatives to the goal, especially in chronic diseases, requiring a lifelong approach.

## 2. The Heme Catabolic Pathway and Metabolic Checkpoints

Bilirubin (BR) and its precursor biliverdin (BV) are entirely derived from heme degradation of hemoglobin (Figure 1), which is catabolized by the unique and ubiquitous microsomal enzyme heme oxygenase (HMOX) [7]. Simultaneously, in the reaction catalyzed by HMOX, a molecule of carbon monoxide (CO), a vasoactive and biologically important gaseous molecule, is released together with iron [8]. BV is not present in the circulation due to its rapid conversion to BR by biliverdin reductase (BLVR), a biologically potent enzyme with an important impact on cell function and metabolism [9]. Interestingly, BR has been shown to have even more versatile biological functions in all organs and compartments of the human body, including general antioxidant activities, immunosuppressive functions, and potent cell signaling properties [1,10,11,12,13,14]. HMOX, the key enzyme responsible for the initiation of heme catabolism, has two isoforms (HMOX1, OMIM 141250; and HMOX2, OMIM 141251) [15,16]. HMOX1 is considered the most inducible enzyme in the human body [17,18], and a variety of natural compounds are known as potent inducers of HMOX1 activity [3,19].

The HMOX pathway, through its activation role in gene transcriptions, is closely associated with another evolutionarily conserved cell system, the multifunctional nuclear factor erythroid 2-related factor (NRF2). It is considered not only a regulator of cellular resistance to oxidants, inflammatory stimuli, and toxic xenobiotics, but also a potent modulator of longevity [20,21] and cardiovascular and metabolic diseases [22]. On the other hand, its aberrant activation can increase the risk of development of various diseases, such as diabetes or cancer [23]. Interestingly, also the NRF2 pathway is highly responsive to natural products (see Table 1). Indeed, both stimulatory and inhibitory effects of a wide array of natural products have been reported, with a possible deep impact on the prevention of various civilization diseases [23,24,25,26,27,28,29], including neurodegenerative conditions, such as Alzheimer’s disease (AD), Parkinson’s disease (PD), schizophrenia, multiple sclerosis, amyotrophic lateral sclerosis, perinatal brain injuries, and Duchenne muscular dystrophy, as well as ischemic stroke, CNS trauma, cerebral neoplasms, etc. [5,30,31,32,33,34,35,36,37,38,39,40]. HMOX1 and its products are believed to protect astrocytes and microglia from increased oxidative stress, apoptosis, and inflammation [35,41,42,43] and to promote angiogenesis [44]. The induction of HMOX1 activity by various natural products has become an important therapeutic target for combating certain neurodegenerative and other autoimmune diseases [19,28,45,46,47,48].

It should be mentioned that, after HMOX1 induction, iron is also released [93]. Especially in the brain, excessive iron levels can be toxic due to its pro-oxidant capacity (as example [5,6,44,94,95,96]). On the contrary, HMOX1, iron, cellular redox status, and inflammation regulate the increased vulnerability to ferroptosis in glioblastoma, the most recurrent brain tumor [97,98,99,100,101,102,103].

Also interesting is the observation that BV enhances the expression of CD36 [104], which is involved in fatty acid oxidation and diabetes control [105]. In this regard, it is interesting to note that CD36 has been identified as the aryl hydrocarbon receptor (AhR) target gene [106], with BR being a potent AhR activator [107]. In fact, almost 400 peroxisome proliferator-activated receptors (PPARs) α-dependent genes have been reported to have been significantly modulated by exposure of BV to HepG2 human hepatoblastoma cells [108]. BV also reduces the activation of c-Jun NH2 terminal kinase (JNK), protecting endothelial cells from undergoing apoptosis in vascular injury-induced intimal hyperplasia [109], and has been reported to be protective in cerebral infarction and cerebral ischemia-reperfusion [110,111]. BLVRA activity has significant implications in AD. Its reduction in AD models not only led to increased BACE1 (β-site amyloid precursor protein cleaving enzyme 1) phosphorylation, resulting in higher Aβ deposits, but it also induces insulin resistance by downregulating the insulin receptor (IR) and inhibiting insulin receptor substrate 1 (IRS1) [112]. Meanwhile, the increase of the BLVRA protein level and activity in the brain of the AD model, followed by the increase of UCB, shows a negative (protective) correlation with oxidative stress markers and cognition [113]. BLVRA, which also functions as a transcription factor, interacts with nuclear factor kappa B (NFκB) and leads to the halt of the cell cycle [114]. This results in the downregulation of BLVRA in brain tumors, specifically, meningiomas and gliomas, with implications not yet elucidated with antioxidant status and chemoresistance in these tumor types [6,115]. Moreover, NFkB activation during hypoxic-ischemic injury [116] may contribute to the underlying cause of cerebral palsy [117,118,119,120] and autism spectrum disorders [121,122,123,124,125].

Bilirubin UDP-glucuronosyltransferase (UGT1A1), a phylogenetically old bio-transforming enzyme conjugating bilirubin with glucuronic acid in the liver cell [7], is another important enzyme in the heme catabolic pathway, which is modulated by various natural products [3,126]. *UGT1A1 *expression is substantially under the control of multifunctional nuclear receptors, including the constitutive androstane receptor (CAR), the pregnane X receptor (PXR), the glucocorticoid receptor (GR), the aryl hydrocarbon receptor (AhR), and the hepatocyte nuclear receptor 1α (HNF1α) [127], which regulate *UGT1A1* transcription via the phenobarbital-responsive enhancer module (PBREM) [127].

Interestingly, BR serves as a ligand for these nuclear receptors, which also contribute to glucose and lipid metabolism [12] and the pathogenesis of cardiovascular diseases and other diseases of civilization [128,129]. Meanwhile, a slightly elevated TSB is negatively correlated with multiple neurologic conditions (for a review on the correlation between TSB and neurologic conditions, see [6]). This strongly supports the idea that the nutraceuticals able to increase bilirubin production might help to prevent neurodegenerative damage [44]. Of relevance, BR has been recently demonstrated to prevent dopaminergic neuron (DOPAn) demise in an ex vivo model of Parkinson’s disease by normalizing TNFα, the determinant in DOPAn death [130]. Remarkably, CAR/PXR/GR/AhR/HNF1α are substantially modulated by dietary components, including natural products [131,132]. It is also important to note that BR per se also serves as a ligand for other metabolic nuclear receptors, such as peroxisome proliferator-activated receptors (PPARs) α and γ, being considered master regulators of cellular and whole-body energy homeostasis, with important roles in the pathogenesis of obesity, diabetes mellitus, metabolic syndrome, and aging [12,133,134,135]. The direct activating effect of bilirubin on PPARα has recently been convincingly demonstrated in various experimental studies (reviewed in [74]). In fact, physiologically relevant BR concentrations were capable of activating PPARα with the same magnitude as that of fenofibrate, a clinically used agonist of this nuclear receptor. In addition, a recent in silico analysis revealed that bilirubin resembles structurally the molecule of fenofibrate [39]. Furthermore, acute treatment of mice with bilirubin resulted in increased expression of hepatic PPARα target genes, including fibroblast growth factor 21 (FGF21) [104]. BR also affects FGF21 [80,81], considered a late-acting fed and fasting-state hormone [136], but interestingly, also an insulin signaling pathway [137,138,139,140] (insulin being considered an immediately acting hormone). BR also affects the expression of PPARγ [91,137], another master regulator of adipogenesis and obesity [141]. Although not all available data are consistent, it seems that modulation of the PPARγ pathway is biologically and clinically relevant. It has been reported that PPARγ agonists activated the AMP-activated protein kinase (AMPK) [77,142,143,144], an important cellular energy sensor [145,146]. Increased AMPK phosphorylation has been reported in subjects with benign hyperbilirubinemia (Gilbert’s syndrome) [77]. PPARγ activation of AMPK also led to inhibition of the mammalian target of rapamycin (mTOR signaling), an evolutionarily conserved nutrient-sensing protein kinase that regulates metabolism, aging processes, and overall life span [147,148], as well as dephosphorylation of a downstream factor, S6K [143]. Similar results on inhibition of S6K phosphorylation were also obtained in our study in human fibroblasts exposed to physiological concentrations of BR [14,149]. Interestingly, dysregulation of AMPK and mTOR hyperactivation was reported in BLVRA-deficient mice [84]; and in another study, BV inhibited TLR4 signaling in leukocytes and triggered phosphorylation of mTOR-specific targets, including Akt, PKC, and AMPK [89], suggesting the importance of BV and BR for modulation of these signaling pathways (Figure 2).

The same effects on the activation of AMPK and mTOR inhibition due to BR were also observed in mice with NAFLD treated with CO, another product of the heme catabolic pathway [85]. PPARγ coactivator- α (PGC1α) is a transcription coactivator of PPARγ that plays a central role in the regulation of cellular energy metabolism [150]. It has been shown that hyperbilirubinemic subjects with Gilbert’s syndrome have substantially lower BMI and serum concentrations of glucose, insulin, C-peptide, and triacylglycerol, the activation of AMPK, PPARα/γ, and peroxisome proliferator-activated receptor-gamma coactivator 1α (PgC1α) being considered the most important factors responsible for these observations [77]. Sirtuins (SIRTs) are a family of signaling proteins involved in metabolic regulation [151]; SIRT1, a deacetylase modulating PPARγ and PGC1α, hence controlling fat and glucose metabolism, is metabolically one of the most important. SIRT1, PGC1α, together with AMPK form an energy-sensing network that controls energy expenditure [152]. Interestingly, BR was demonstrated to upregulate SIRT1 in an experimental NAFLD in vitro study [87]. Furthermore, SIRT1 influences the acetylation of microtubule-associated tau (MAPT) that contributes to the preservation of neuronal cytoskeletal stability important for neuroprotection in cerebral ischemia/reperfusion [153]. All BR-modulated pathways described above are interlinked, as demonstrated by the role of the CD39 ectonucleotidase/adenosine pathway in immunity and inflammation, which is under the control of HMOX1 [154,155], and also have important effects on glucose metabolism and insulin signaling [156], the AMPK pathway [157], and the pathogenesis of liver diseases, such as metabolic dysfunction-associated fatty liver disease (MAFLD) [158,159], indicating the complexity of BR-related modulation of cell signaling. It should be noted that some of the anti-inflammatory effects of CO are mediated via its CD39 ectonucleotidase/adenosine pathway [79]. All of these experimental data fit in with the clinical observation of low PPARα expression, as well as systemic BR concentrations in obese subjects [160]. In addition, BR has also been shown to ameliorate experimental colitis by upregulation of CD39 [78]. The effect of the heme catabolic pathway on cell metabolism is probably even more complex. As proven in a liver-specific BLVRA knockout mouse model [91], BLVRA (and hence bilirubin, its enzymatic product) regulates hepatic lipid metabolism by directly affecting the key enzymes implicated in lipid metabolism. In addition to AMPK, these also include acetyl-CoA carboxylase, an essential and rate-limiting enzyme in fatty acid metabolism [161] and glycogen synthase kinase 3b (GSK3b), one of the most active cellular kinases, with more than 100 known targets, involved in the regulation of multiple cellular functions, including lipid and glucose metabolism, among others [162,163]. Thus, BR appears to act as a multifunctional modulator at multiple cellular metabolic checkpoints, which are often mutually interrelated. This is exemplified by bilirubin-induced suppression of p38 MAPK (but also other MAPKs) [164] in ischemia/reperfusion after heart transplantation, known to be crucial for insulin signaling, as well as atherogenesis [165].

Several natural products have been reported to activate both PPARα and PPARγ nuclear receptors [75,76], but also AMPK [90]. Similarly, a variety of natural products, including curcumin, resveratrol, quercetin, or apigenin, to mention at least some of them, have been shown to inhibit mTOR signaling [86] or activate SIRT1 [88] (summarized in Table 1).

## 3. Natural Compounds with Demonstrated Effects on Bilirubin and Its Metabolic Enzymes

### 3.1. Natural Compounds

#### 3.1.1. Flavonoids

There is strong clinical evidence indicating that treatment with silymarin (a seed extract of milk thistle, *Silybium marianum *(L.) Gartn.) [166] flavonolignans results in mild elevation of TSB concentrations, as reported in silybin-treated patients with prostate cancer [167], as well as hepatitis C [168]. It is important to note that some of the minor silymarin flavonolignans are even more biologically active, as evidenced in our recent experimental study, in which dehydrosilybin A and B were potent partial inhibitors of UGT1A1 activity [166]. Flavonoids are also promising chemoadjuvants for the treatment of neurodegenerative diseases [169]. Fisetin is one of the examples of flavonoid tested with positive results in the experimental model of amyotrophic lateral sclerosis by inducing HMOX1 [170]. A natural aglycone flavonoid from the Erigeron plant, known as breviscapine, decreases neuronal apoptosis by promoting the expression of NRF2, followed by the increase in antioxidant enzymes, including HMOX1, in traumatic brain injury [54]. Eriodictyol, a flavonoid compound that is commonly present in the rinds of citrus fruits and certain traditional Chinese herbal remedies, also exhibits anti-AD’s effect. It ameliorates memory impairment, inhibits Aβ aggregation, and decreases Tau phosphorylation in the brains of AD mice by modulating the Nrf2/HMOX1 signaling pathway [171].

Another compound in this category, luteolin, has also been proposed as a phototherapeutic agent for PD due to its antioxidant, anti-inflammatory, and anti-apoptosis effects. It can enhance the HMOX1 expression and the link between Nrf2 and its antioxidant element, providing a good adaptive survival response against oxidative damage. Furthermore, this compound also inhibits the expression of various pro-inflammatory compounds, after the LPS stimulus, such as inducible NO synthase (iNOS), TNFα, cyclooxygenase 2 (COX2), interleukin 1β (1Lβ), nitric oxide (NO), and prostaglandin E2 [60].

#### 3.1.2. Curcumin

Curcumin is a major polyphenolic compound from the curcuminoid group of phytochemicals, originating from the plant *Curcuma longa*. Curcumin is a potent inducer of HMOX1 with many clinical consequences. Its use as a nutraceutical has been investigated in several studies focused on the prevention of neurotoxicity and neurodegeneration (reviewed in [172]). Despite the small number of participants, a clinical trial of a curcumin formulation with enhanced oral bioavailability showed an improvement in survival probability over a 12-month period in amyotrophic lateral sclerosis [173,174]. In an in vitro model of the disease (microglia cells that have been activated by LPS), curcumin reduces the production of iNOS and NO release by increasing the expression of NRF2 and HMOX1 and downregulating the NFkB signaling pathway, which inhibits the release of pro-inflammatory cytokines IL6, IL1, and TNFα [175]. NFkβ/MAPK pathways have also been involved in curcumin-mediated anti-inflammatory action in microglia [61]. Similarly, in primary astrocytes, curcumin activates NRF2 target genes, including HMOX1 and NQO1 (NAD(P)H dehydrogenase quinone 1); lowers the amount of intracellular ROS; and lessens oxidative damage and mitochondrial dysfunction [176], as well as curcumin; it also induces an improvement of mercury-induced cytotoxicity via the NRF2/ARE/PKCδ pathway [64]. The antioxidant properties of curcumin have also been shown in neurons, by a NRF2/PKCδ-mediated induction of p62 (ubiquitin sensor and selective autophagy receptor) phosphorylation [70]. Although the in vitro neuroprotective action of curcumin (and astragaloside) in depression has been suggested to be mediated by the tyrosine protein kinase (TRK) β/MAPK/PI3K/CREB signaling pathways-induced upregulation of BDNF (brain-derived neurotrophic factor) [177], in in vivo models, the antidepressant properties of curcumin have been found to be involved via AKT1, NRF2, and ARE signaling [65,71]. Furthermore, in vivo, dietary supplementation with curcumin has been shown to reverse the microglial and astrocytes activation via NRF2/TLR4/NFkB/RAGE (receptor for advance glycation end products) signaling in a rodent model ethanol-induced neurotoxicity [68], and acts as an antioxidant in intracerebral hemorrhage, traumatic brain injury, quinoline acid-induced glutamate neurotoxicity, and ischemic injury models via the NRF2/HMOX1/AKT pathways [49,62,63,66,67,72]. Moreover, curcumin also exhibits anti-cancer effects by enhancing HMOX1 expression and activating ferroptosis, a form of oxidative cell death, in thyroid cancer [178].

#### 3.1.3. Astragaloside

Astragaloside IV (AST) is one of the main active ingredients of *Astragalus* membranaceus var. mongholicus, or A. membranaceus, a traditional Chinese herb. AST was found to exhibit neuroprotection in rats with ischemic stroke by upregulating SIRT1 expression, which promotes anti-inflammation and antioxidant production [153]. Similarly, AST reduces LPS-mediated neuroinflammation (microglia) by acting on the activation of NRF2, and HMOX1 via the ERK pathway [57]. AST also improves endothelial dysfunction in cardiovascular disease by activating the NRF2 signaling pathway and promoting the transcription of antioxidant enzymes including HMOX1 [179]. The combination of AST and Panax notoginseng results in protection in cerebral ischemia-reperfusion mice by increasing the nuclear translocation of Nrf2, followed by the increase of HMOX1, with a decrease of oxidative stress markers and an increase of the SOD and GSH level [180].

#### 3.1.4. Vitamins

Vitamin C consumption in combination with vitamin E for one month among AD patients could reduce oxidative stress markers in plasma and CSF (cerebrospinal fluid) [181]. Vitamin C indeed promotes antioxidant activity by activating NRF2 and increasing HMOX1 expressions [182]. It also acts against inflammation by decreasing TNFα and IL6 and increasing IL10 [183]. There is growing evidence that vitamin D deficiency can affect brain processes, including cognition, and significantly increase the risk of AD [184], speeding up the aging process through the disruption of redox cell signaling pathways [184]. In line, maxacalcitol, a vitamin D analogue, substantially lowered neuroinflammation and enhanced expression of NRF2 and its downstream effectors (HMOX1 and GSH) in the animal model for AD [185].

#### 3.1.5. Madecassoside

The Indian Ayurvedic system of medicine has utilized *Centella asiatica* to improve neurocognitive functioning. The most abundant triterpenoid saponin isolated from the plant, madecassoside, protects the neurons from apoptosis due to free radicals by increasing the antioxidant activity in the ALS mouse model. It has also been reported to improve LPS-mediated neurotoxicity in rats with its anti-inflammatory activities and activation of the Keap1-Nrf2/HMOX1 signaling pathway [186].

#### 3.1.6. Green Tea

The clinical study by Arab et al. showed that drinking green tea (leaf of *Camellia sinensis*) 2 g/day for two months has a positive impact on cognitive performance in severe AD patients by increasing the total antioxidant capacity by around 20% [187]. Additionally, epigallocatechin-3-gallate (EGCG), a bioactive compound of green tea, has been demonstrated to stimulate antioxidant defense mechanisms by activating the Nrf2/ARE pathway and antioxidant enzymes by activating AKT and ERK1/2 pathways [188]. In particular, EGCG functions as an antioxidant by modulating neurodegenerative inflammatory processes, such as ferroptosis and microglia-induced cytotoxicity, and by activating signaling pathways important for neuronal survival [174].

#### 3.1.7. S-Allyl Cysteine

S-allyl cysteine, a water-soluble organosulfur compound containing an amino acid isolated from the garlic bulb (*Allium sativum*), ameliorates cognitive deficits in streptozotocin-diabetic rats via suppression of oxidative stress, inflammation acting on the NRF2- HMOX1, TLR4, and NFκB signaling cascade [58].

#### 3.1.8. 20C

20C, a bibenzyl compound isolated from *Gastrodia elata *(a commonly used traditional Chinese medicine for therapeutic applications, like epilepsy and vertigo treatment), protects PC12 neurons from rotenone-induced lesions mimicking PD in vitro by normalizing (increasing) HMOX1 and NQO1 mRNA and the protein expression level, and by reverting ROS accumulation, cytochrome c release, and apoptosis via NRF2 signaling [47].

#### 3.1.9. *Achyranthes bidentata*

The extract *Achyrantes bidentata* polypeptide K (ABPK) from the *Amaranthaceae *family, another Chinese herbal medicine, reduces inflammation in LPS-challenged microglia (BV2 cells, NO, IL6, TNFα, PGF2) via NRF2/HMOX1 signaling [43].

#### 3.1.10. *Coriolus versicolor* and *Hericium erinaceus*

Consumption of certain mushrooms affects neurocognitive functioning. Two of the widely studied medicinal mushrooms are *Coriolus versicolor *and *Hericium erinaceus. *The combined extracts of these two mushrooms lower LXA4, a metabolic product of arachidonic acid, which is considered an endogenous “stop signal” for inflammation, astrocytes and microglia activation, α-synuclein content, apoptosis, and dopaminergic neuron death, finally improving the motor abilities in a rodent model of PD [48].

#### 3.1.11. Hyperoside (Quercetin 3-O-galactoside)

Hyperoside (quercetin 3-O-galactoside) from *Acer tegmentosum*, a Korean traditional medicine, has demonstrated antioxidant action in cultured neurons serving as an in vitro model of PD via NRF2/HMOX1 [52].

#### 3.1.12. Acerogin A

Acerogenin A, a natural compound from *Acer nikoense* Maxim used in Japanese folk medicine, has been shown to prevent glutamate neurotoxicity in vitro by acting through NRF2/HMOX1 and PIK3/Akt signaling, with glutamate neurotoxicity being a common pathological mechanism in many neurodegenerative and neurologic conditions [55].

#### 3.1.13. Kaempferol, Ginsenoside rh2

Kaempferol and ginsenoside rh2, the most active principle of the Kaixinsan formula, used as an antidepressant in traditional Chinese medicine, are effective in reverting the pro-oxidant cellular status of SH-SY5Y cells exposed to H_2_O_2_ by increasing thioredoxin reductase activity via NRF2/AKT [71].

#### 3.1.14. Mangiferin

Mangiferin is a natural compound originating from multiple plants, including *Mangifera indica *L. Through recent studies, it has been demonstrated to have an important role in protecting neurons from degeneration. In fact, it has some crucial antioxidant properties by preventing the formation of hydroxyl radicals and ROS [59]. Mangiferin has protective effects on PD in vitro and in vivo models by enhancing antioxidant defense, including the expression of NRF2 and HMOX1 [189]. Mangiferin also can downregulate NFκB in breast cancer, followed by increased apoptosis as one of the consequential effects [190].

#### 3.1.15. Ellagic Acid

Ellagic acid is a chromene-dione derivative (C_14_H_6_O_8_) present in many fruits and nuts, such as pomegranates, black raspberries, raspberries, strawberries, walnuts, and almonds. It is a phenolic antioxidant with higher antioxidant activity than vitamin E, succinate, and melatonin. It can also inhibit the major pro-inflammatory pathways, such as NFkB, MPAKs, and JAK/STAT, preventing inflammatory molecules’ release (e.g., TNFα, IL1β, IL6, and iNOX) [69].

### 3.2. Natural Compounds Mimicking Products of the Heme Catabolic Pathway

#### 3.2.1. Tetrapyrroles from *Spirulina platensis*, Phycocyanin, and Phycocyanobilin

*Spirulina (Arthrospira) platensis* is a blue-green freshwater alga widely used as a dietary supplement. It is rich in proteins, carotenoids, essential fatty acids, vitamin B complex, vitamin E, and minerals, such as copper, manganese, magnesium, iron, selenium, and zinc. It is a source of potent antioxidants, including spirulans (sulfated polysaccharides), seleno compounds, phenolic compounds, and phycobiliproteins (C-phycocyanin and allophycocyanin) [191]. Numerous studies have demonstrated that dietary supplementation of *S. platensis* is helpful in the prevention and treatment of atherosclerosis, diabetes, and/or cancers (for review see [192]). C-phycocyanin is a light-harvesting biliprotein that is possibly implicated in the biological effects of *S. platensis*. C-phycocyanin contains a covalently linked chromophore called phycocyanobilin (PCB) [193], a linear tetrapyrrolic molecule structurally similar to BV that constitutes up to 1% of the dry weight of *Spirulina*. Interestingly, phycocyanobilin (PCB) can be metabolized by BLVR to phycocyanorubin, similarly as BV is reduced to BR in the human body [194]. Therefore, *S. platensis* appears to be a rich nutraceutical source of PCB [195], which is known to significantly inhibit nicotinamide adenine dinucleotide phosphate (NADPH) oxidase activity, activate the action of antioxidant enzymes, and have anti-inflammatory and cell signaling activities with an expected substantial impact on human health, including fighting the neurologic adverse effects of COVID-19 [73,196,197].

We have shown that *S. platensis* and PCB markedly increased Hmox1 in experimental models of atherosclerosis [198]. Therefore, activation of HMOX1 and the heme catabolic pathway appears to be an important mechanism of this food supplement for the reduction of atherosclerotic diseases. Due to their antioxidant effects, C-phycocyanin and PCB (but also BR and BV) protected diabetic db/db mice against albuminuria and renal mesangial expansion in db/db mice, and normalized tumor growth factor-β and fibronectin expression, suggesting a novel and feasible therapeutic approach to prevent diabetic nephropathy [199]. Importantly, antidiabetic effects of *S. platensis*, C-phycocyanin, and PCB were reported in diabetic male albino rats treated with streptozotocin, with decreases of blood glucose concentrations, improvement of insulin resistance and blood lipids levels, and restoration of pancreatic cell morphology [200].

The anticancer effects of *S. platensis* and its tetrapyrrolic components (PCB and chlorophyllin, a surrogate molecule for chlorophyll A) in biologically relevant doses were proved on experimental pancreatic cancer models in another study. These models included nude mice xenotransplanted with pancreatic cancer cells, suggesting a chemopreventive role of this edible alga, whose dietary supplementation with this alga might enhance the systemic pool of linear tetrapyrroles mimicking BR [201]. Similarly, in an experimental in vitro study by Hussein et al., C-phycocyanin and PCB treatment led to potent antiproliferative, and pro-apoptotic effects in MCF-7 breast cancer cells [202].

PCB also showed strong anti-inflammatory and general hepatoprotective effects in mice with CCl4-induced liver injury, with a marked improvement inf the survival rate of acute liver failure in mice injected with a lethal dose of CCl4 [203].

Importantly, *Spirulina alga* and its associated bioactive compounds appear to also be important in chemoprevention of neurodegenerative diseases [73,204]. Activated microglia, displaying NADPH oxidase activity, are believed to contribute substantially to the pathogenesis of many brain diseases, such as PD and AD, and multiple sclerosis, but also cerebral ischemia [205,206,207] or COVID-19-induced damage of the nervous system [73]. Although direct clinical evidence is lacking, experimental data suggest that *S. platensis* extracts could ameliorate the risks of these diseases [208]. A diet high in *Spirulina* ameliorates the loss of dopaminergic neurons in the mouse model of PD [209]. Furthermore, PCB improved the neurological outcomes in a mouse model of experimental autoimmune encephalitis (EAE) that mimics multiple sclerosis conditions, with significant inhibitory effects on pro-inflammatory cytokines. Interestingly, a reduction of demyelination, active microglia/macrophages density, and axonal damage was detected, along with an increase in oligodendrocyte precursor cells and mature oligodendrocytes, when assessing the spinal cords of EAE mice treated with PCB. Due to the increased treatment effects of PCB when used with IFNβ therapy, PCB was suggested for use with IFNβ as a disease-modifying agent for multiple sclerosis [210]. Similar results were found also in other EAE experimental studies, demonstrating a clear tendency for amelioration of the clinical severity of the disease promoted by the treatment with PCB in an EAE model with a reduction in the levels of the pro-inflammatory cytokines IL17, IFNγ, and IL-6 in the brains of animals treated with PCB. Similar observations were also obtained in animal models of cerebral ischemia [205,211]. The demyelination potential of C-phycocyanin and PCB with their possible clinical use in patients with demyelinization disorders has been reviewed recently [207].

PCB also was shown to have potential inhibitor activity against main protease (Mpro) and papain-like protease (PLpro) of human and animal coronaviruses, indicating broad-spectrum inhibitor activity of PCB. In addition, in silico studies with the Mpro and PLpro enzymes revealed that other tetrapyrrolic phycobilins, such as phycourobilin and phycoerythrobilin, will have similar binding affinity to SARS-CoV-2 Mpro and PLpro [212]. Hence, PCB and *S. platensis* were suggested as promising nutraceuticals against COVID-19-induced brain damage [73]. In addition, antiviral activities of *S. platensis-*derived compounds were shown in a study by Chen et al., who reported inhibitory effects of Spirulina extract against influenza virus replication and reduction of virus-induced mortality [213]. In in vitro studies, antiviral activities of *S. platensis *extract were also reported against herpetic viruses, measles, and mumps viruses, as well as human immunodeficiency virus (for a review, see [214]). Interestingly, in a small clinical study, *S. platensis* consumption for 12 months improved the immunological profile of HIV-infected patients [215]. It should be noted that in countries with high algae consumption, such as Japan, Korea, or Chad, the prevalence rate of HIV infection and AIDS remains substantially lower compared to other countries [214,216].

Due to the potent antioxidant, antibacterial, and other beneficial biological effects, extracts from *S. platensis* exhibits have been used in skincare formulations for the treatment of acne [217], sunscreens (*S. platensis* having a very high sun protecting factor, SPF) [218], face masks lip balms, and ointments for their wound-healing and anti-aging properties [219]. Therefore, the global market is estimated to be as much as USD 2000 million by 2026, with an overall *S. platensis* consumption of more than 321,000 tons [220].

#### 3.2.2. Artificial Bezoar

BR is present also in pulverized bovine gallstones (*Calculus bovis*, artificial bezoar, *Niu Huang* in Chinese) used for centuries for a variety of human diseases in traditional Chinese medicine [221,222]. Preparations of these pulverized gallstones have a high content of BR of no less than 25% by weight [222].

Neuroprotective effects of artificial bezoar were reported in an experimental study on male Sprague-Dawley rats with induced cerebral ischemia. Pretreatment with An-Gong-Niu-Huang Wan, a complex traditional Chinese medicine formulation containing Calculus bovis, significantly ameliorated ischemic damage to the brain in a dose-dependent manner, including a reduction in the neurological deficit score and infarct area [223]. Other reported activities of *Calculus bovis* on the nervous system include sedative, anticonvulsant, analgesic, antiepileptic, or even anti-schizophrenic (reviewed in [222]). Furthermore, *Calculus bovi*s was found to also have protective effects on the heart, vessels, lungs, or liver [222], or even experimental cancer, as shown for experimental breast cancer, human hepatoblastoma [224].

#### 3.2.3. Chlorophylls

Chlorophyll is a tetrapyrrolic compound structurally related to BR [7] that is likely to exert similar protective biological activities. Dietary intake of green leafy vegetables rich in chlorophyll is associated with protection against cancer and other civilization diseases, including neurodegenerative diseases [50,225]. Although their resorption from the gut lumen is not high, chlorophylls appear to be important for their potential systemic cancer-preventive effects [226], reaching biologically relevant concentrations even in peripheral tissues [227]. Anti-proliferative and antioxidant effects of chlorophylls (chlorophyll a/b, chlorophyllin, and pheophytin a) were reported recently in our experimental in vitro and in vivo pancreatic cancer study [228].

## 4. Natural Compounds Targeting the Heme Catabolic Pathway

As described in detail in our recent review [14], targeting the individual parts of the heme catabolic pathway represents an important way to increase tissue or systemic levels of BR.

### 4.1. Modulation of HMOX1

As emphasized above, HMOX1 is inducible by a variety of natural products and is an important chemotherapeutic target for the prevention and treatment of many neurodegenerative and other autoimmune diseases [19,45,46]. These natural HMOX1 modulators include various polyphenolic compounds found in plants, such as curcuminoids; flavonoids, such as quercetin, EGCC, genistein, eriodictyol, or certain flavonolignans of silymarin complex, caffeic acid, and resveratrol; and natural coumarins, such as esculetin or fraxetin [3,229,230].

### 4.2. Modulation of BLVRA

BLVRA induction seems always beneficial, while its enzymatic inactivation looks detrimental, possibly by reducing the final concentration of unconjugated bilirubin (UCB) inside the cell [9]. Interestingly, BLVRA expression is also inducible by natural products, leading to increased production of BR, as demonstrated for Korean red ginseng in an in vitro study in murine hippocampal astrocytes improving mitochondrial functions via the LKB1-SIRT1-ERRα axis [231].

### 4.3. Modulation of the Hepatic Transport of Bilirubin

Inhibition of organic anion transporter 1B1 (OATP1B1), a transporter crucial for UCB uptake in the liver, by compounds or substances relying on this transporter for transport could lead to a substantial increase in bilirubin concentrations [3]. Basolateral uptake of bilirubin in liver cells is another druggable target to increase mild serum BR concentrations. In fact, interference with OATP1B1 bilirubin transporter was reported for many natural compounds commonly present in certain foods, food supplements, and herbs [3,232].

### 4.4. UGT1A1 Modulation

As UGT1A1 is responsible for intrahepatic BR conjugation, its suppression will lead to an increase in BR level [3]. Many natural compounds, often used as nutraceuticals, have UGT1A1-suppressing activity resulting in mild elevation of systemic BR concentrations, including silymarin flavonolignans or EGCC, or many Japanese and Chinese herbs commonly used in traditional herbal medicine [3].

### 4.5. Gut Microbiome Modulation

By affecting the enterohepatic and enterosystemic circulation of BR [233], BR-reducing bacteria in the gut lumen can affect systemic concentrations of the pigment [234]. Although no experimental and clinical data on possible modulation of gut microbiome metabolizing bilirubin are available today, likely, certain probiotics or other modulating agents capable of affecting intestinal metabolism of bilirubin with possible beneficial clinical consequences will become available [235].

## 5. Conclusions

The beneficial potential of bilirubin in human health is clear in large measure. The modulability of HMOX1 and its metabolic checkpoints by natural compounds makes this system a great and feasible target to combat various public health concerns and age-related diseases with minimal changes in the diet. Further studies, and collaborations with the food or pharma-nutraceutical industry, are needed to specify all this solid information in a varied and easy-to-take functionalized alimentary regimen to promote health day by day.

## Figures and Tables

**Figure 1 biomolecules-14-00063-f001:**
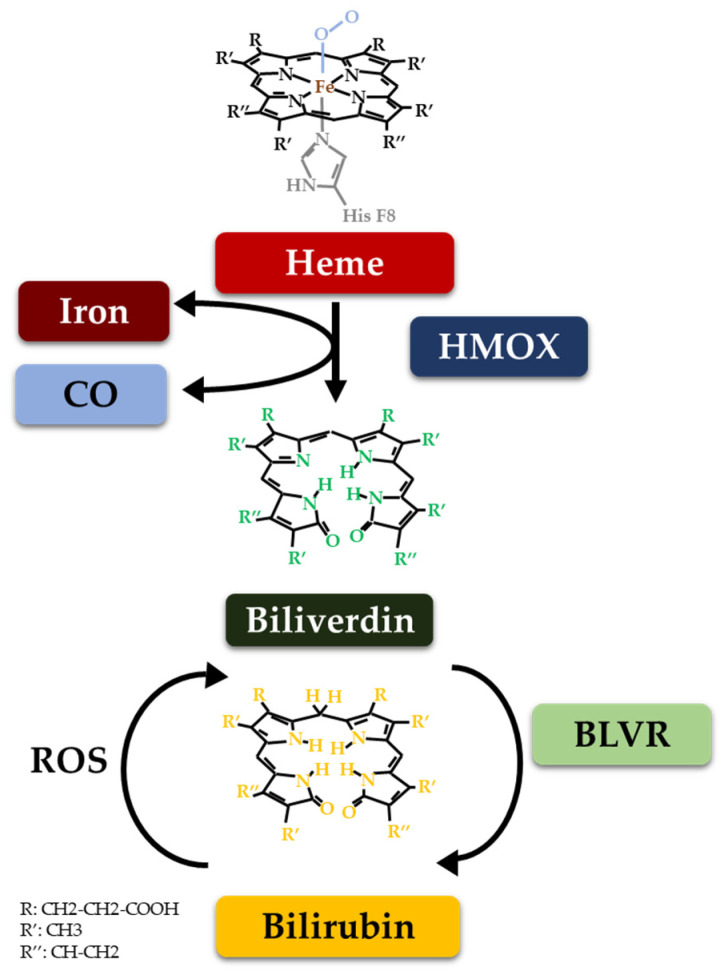
Bilirubin metabolism. CO: carbon monoxide, HMOX: heme oxygenase, ROS: reactive oxygen species, BLVR: biliverdin reductase.

**Figure 2 biomolecules-14-00063-f002:**
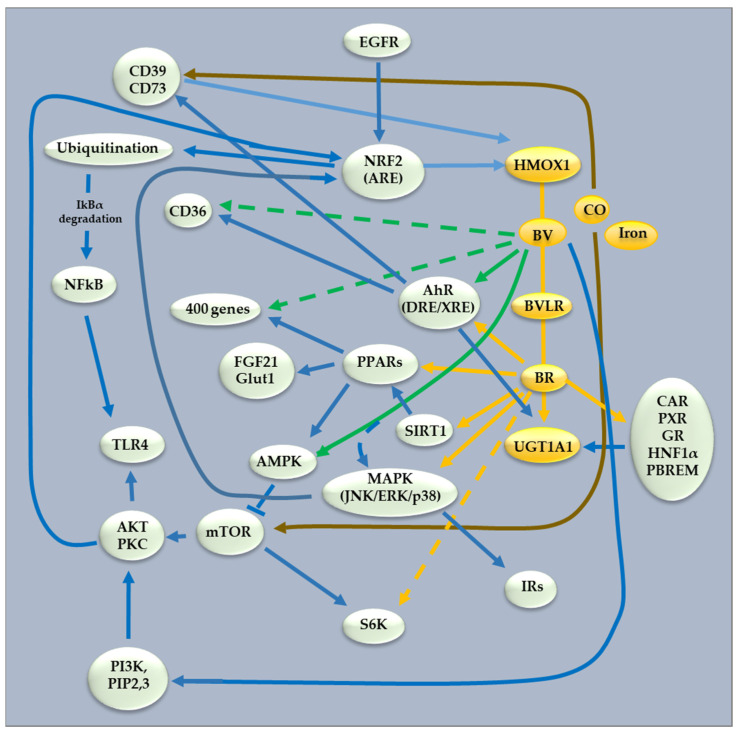
Interconnections between heme metabolism and signaling pathways. Solid line: experimental evidence of a direct link. Dash lines: no experimental evidence of a direct action. Green lines: biliverdin action; yellow lines: bilirubin actinin; brown lines: CO action; blue lines: all other actions/connections. Yellow EFGR: epidermal growth factor receptor; NRF2: nuclear factor erythroid 2–related factor 2; ARE: antioxidant response element; CD: cluster of differentiation; HMOX: heme oxygenase; BV: biliverdin; BVLR: biliverdin reductase; BR: bilirubin; UGT1A1: UDP-glucuronosyltransferase 1A1; CAR: constitutive active/androstane receptor; PXR: pregnane X receptor; GR: glucocorticoid receptor; AhR: aryl hydrocarbon receptor; DRE/XRE: drought/xenobiotic-responsive elements; HNF1: hepatocyte nuclear factor 1α; PBREM: phenobarbital (PB)-responsive enhancer module; PPAR: peroxisome proliferator-activated receptor; SIRT: silent information regulator; MAPK: mitogen-activated protein kinase; JNK: Jun kinase; ERK: extracellular signal-regulated kinase; mTOR: mechanistic Target of Rapamycin; AMPK: AMP-activated protein kinase; AKT: protein kinase B; PKC: protein kinase C; TLR4: toll-Like receptor 4; NFkB: nuclear factor-κB; PIP: phosphatidylinositol biphosphate; PI3K: phosphatidylinositol-3-kinase; IRs: insulin receptor; S6K: protein S6 kinase; FGF: fibroblast growth factor; Glut1: glucose transporter type 1.

**Table 1 biomolecules-14-00063-t001:** Metabolic checkpoints affected by natural compounds and also by the heme catabolic pathway. NRF2, nuclear factor erythroid 2—related factor 2; HMOX, heme oxygenase; PI3K, phosphoinositide 3-kinase; AKT, serine/threonine kinase; EGFR, epidermal growth factor receptor; ERK, extracellular signal-regulated kinase; ROS, reactive oxygen species; NQO1, NAD(P)H: Quinone Acceptor Oxidoreductase Type 1; NO, nitric oxide; TNFα, tumor necrosis factor alpha; IL, interleukin; SOD, superoxide dismutase; GSH, glutathione; PERK, protein kinase R (PKR)-like endoplasmic reticulum kinase; NFkB, nuclear factor k-light-chain-enhancer of activated B cells; TLR4, toll like receptor 4; MAPK, mitogen-activated protein kinases; PKC, protein kinase C; GST, glutathione g-transferases; p62, selective autophagy receptor p62; AKT, protein kinase B (PKB), also known as Akt; NADPH, nicotinamide adenine dinucleotide phosphate; PPAR, peroxisome proliferator-activated receptor; PGC1α, peroxisome proliferator-activated receptor-α coactivator-1α; CD, cluster of differentiation; CO, carbon monoxide; FGF21, fibroblast growth hormone 21; mTOR, mammalian target of rapamycin; SIRT1, NAD-dependent deacetylase sirtuin-1; AMPK, AMP-activated protein kinase; T2DM, type 2 diabetes; GSK3b, glycogen synthase kinase-3b; BLVRA, biliverdin reductase A.

Metabolic Checkpoint	Heme Catabolic Pathway Modulator	Natural Compound (Some Examples)	Possible Clinical Impact
NRF2	NRF2 activates HMOX1 [29,49,50] Bilirubin activates NRF2 [51]	Sulphoraphan, curcumin, bixin, apigenin, cinnamaldehyde, withaferin A, luteolin, wogonin, chrysin… [23,24,25,26,27,28]	Regulator of cellular resistance to oxidants, inflammatory stimuli and toxic xenobiotics, modulator of longevity and cardiovascular and metabolic diseases.
	HMOX1	20C (bibenzyl compound isolated from Gastrodia elvata) [47]	Suppresses the pro-apoptotic effect of Rot by inhibition of Bax and suppress the accumulation of intracellular ROS and the collapse of the mitochondrial membrane potential.
	HMOX1	(ABPK) achyrantes bidentata polypeptide K [43]	Neuroprotective agent inhibiting the neuroinflammation on BV2 microglia cell culture.
	HMOX1	Coriolus versicolor, Hericium erinaceus [48]	Anti-inflammatory modulating the lipoxin A4 levels (LXA4), resolving neuroinflammation and limiting the motor and non-motor symptoms, typical of PD.
	HMOX1	Hyperoside (quercetin 3-O-galactoside) [52]	Protects cultured dopaminergic neurons from death via ROS-dependent mechanisms.
	HMOX1	Berberine (BBR) [53]	Binds specific DNA sequences triggering DNA repair process.
	HMOX1	Breviscapine [54]	HMOX1 and NQO1 increases.
	HMOX1 via PI3K/AKT	Acerogin A [55]	Prevent glutamate-induced oxidative damage.
	HMOX1 via EGFR/ERK	Astragaloside IV+/− Panax notoginseng [56,57]	Reduction of the oxidative stress markers, inhibition inflammatory mediators (NO, TNFα, IL6) and increase of SOD and GSH level.
	HMOX1 and NFkB/TLR4 signaling cascade	S-allyl cysteine (SAC) from aged garlic extract [58]	Improve cognitive deficits by attenuation of oxidative stress and neuroinflammation.
	HMOX1	Mangiferin [59]	Protects neurons and glia from the oxidative damage by increasing HMOX1 in AD.
	HMOX1	Luteolin [60]	Increases cells’ survival by preventing apoptosis and oxidative stress.
	* - *	Curcumin [61]	Inhibits the secretion of pro-neuroinflammatory mediators by increasing HMOX.
	-	Curcumin [62,63,64,65]	Protects neurons by ameliorating brain water content, oxidative stress, inflammation, and apoptosis, as well as reversal of depressive-like behaviors.
	*-*	Quercetin, anthocyanins, tea polyphenols, kaempferol, hesperetin, icariin, and various forms of terpenoids [28]	Protect from glutamate neurotoxicity and rescue of impaired cognitive function by increasing antioxidant responses, improving cell viability, and decreasing pro-inflammatory mediators.
	* - *	Curcumin [66]	Improves motor deficits and morphological alterations through antioxidant activity in an in vivo model of quinolinic acid neurotoxicity.
	NRF2 and PERK pathway	Curcumin [67]	Improves motor, sensory, reflex, and balance through inhibition of oxidative stress and apoptotic process.
NFkB/TLR4	NRF2 and NFkB	Curcumin [68]	Improves memory and behavior.
NFkB/STAT3/Ap-1		Luteolin [60]	Reduces neuroinflammation induced by astrocytes.
NFkB/MAPKs	NFkB/MAPK pathways	Curcumin [61]	Inhibits the secretion of pro-neuroinflammatory mediators by increasing Hmox.
NFkB		Ellagic acid [69]	Promotes anti-inflammatory and anti-antioxidant effects in AD and PD.
PKC	PKC activates NRF2	Curcumin [70]	Neurons are stimulated to increase antioxidant gene expression (GST-mu1, NQO1, and Hmox1), as well as p62, resulting in a positive feedback loop.
ERK	ERK modulate NRF2 anti-oxidant signaling	Curcumin [66]	Improve motor deficits and morphological alterations through antioxidant activity.
	AKT2/NRF2 pathways	Kaempferol Ginsenoside rh2 [71]	Upregulation of the antioxidant enzyme thioredoxin linked to antidepressant mechanism.
	AKT/NRF2	Curcumin [72]	Protects neurons and reduces infarct size in in vitro (oxygen and glucose deprivation/reoxygenation) and in vivo (middle cerebral artery occlusion) models of ischemic injury.
NADPH oxidase	Bilirubin, biliverdin	C-phycocyanin (C- PC) [73]	Protective in many neurodegenerative diseases and in COVID-19-induced neurologic damage.
PPARs	Bilirubin [74]	Resveratrol [75,76]	Beneficiary effects on glucose and adipose tissue metabolism.
PGC1a	Bilirubin [77]	Resveratrol, quercetin, curcumin, saponins, epigallocatechin-3-gallate (EGCG) [37]	Regulation of cellular energy metabolism with beneficiary effects on civilization diseases.
CD39	Bilirubin [78] CO [79]	Resveratrol [38] Curcumin [39]	Control of inflammatory processes via purinergic signaling.
FGF21	Bilirubin [80,81]	Coffee phytochemicals (chlorogenic/protocatechuic acid) [82] Cocoa phytochemicals (theobromine/protocatechuic acid) [83]	Energy homeostasis, via FGF21 signaling, a late-acting fed and fasting-state hormone.
mTOR	Bilirubin [10,14] biliverdin [84] CO [85]	Curcumin, quercetin, apigenin [86]	Modulation of nutrient-sensing with impact on intermediary metabolism, aging processes, and overall life span.
SIRT1	Bilirubin [87]	Resveratrol, butein, quercetin [88], astragaloside IV [40]	Control of fat and glucose metabolism, and energy expenditure. Anti-inflammatory and antioxidant. Reducing infarct size in ischemic stroke.
AMPK	Bilirubin [77] biliverdin [89] CO [85]	Resveratrol Berberine Quercetin [90]	Prevention of cardiovascular and metabolic diseases (T2DM), energy homeostasis.
GSK3b	BLVRA/bilirubin [91]	Resveratrol, curcumin, berberine [92]	Modulation of cellular kinase, with >100 known targets affecting lipid and glucose metabolism, and cell proliferation.

## Data Availability

All the data are inside the review.

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
