# Peer review of "Role of Natural Compounds Modulating Heme Catabolic Pathway in Gut, Liver, Cardiovascular, and Brain Diseases"

_biomolecules, 2024, doi:10.3390/biom14010063_

Round 1

Reviewer 1 Report

Comments and Suggestions for Authors

This article by Jayanti et al. provides a helpful overview of natural compounds that target the heme catabolic pathway and modulate bilirubin production. It summarizes current research on how components in foods, herbs and nutritional supplements can interact with enzymes involved in bilirubin metabolism. The range of natural compounds covered is extensive, encompassing flavonoids, curcumin, vitamins, and more. For each substance or class of substances discussed, the known or hypothesized mechanisms of action related to the heme catabolic pathway are explained. However, this manuscript would benefit from some organizational and structural improvements to enhance flow and readability, as outlined below.

Comments:

1)      As the title indicates, this review is intended to cover the impact of natural heme catabolic pathway modulators on common diseases afflicting the “gut, liver, cardiovascular system and brain.” However, while extensive information is provided cataloging natural compounds and their mechanisms of action on bilirubin metabolism, direct discussion applying this to health outcomes is limited.  I recommend adding a new section, or table to help clearly map the associations between promising natural modulators, their mechanisms of action on the heme catabolic pathway, and resulting therapeutic or protective effects on outcomes in relevant disease models. Having a visual summary that establishes connections between compounds, effects on pathways/enzymes, and associated impact on pathology and symptoms for disorders of interest would help fulfill the framing set by the title. 

 2)      The abstract opens by introducing bilirubin (BR) and biliverdin (BV) as key products of the heme catabolic pathway. However, structurally BR and BV originate from the breakdown of heme within cells. Since heme is the initial substrate for this pathway, the abstract would benefit conceptually from introducing heme upfront and briefly explaining its catabolism into BR and BV. This would orient readers by establishing the origin of these downstream molecules and laying the groundwork for how the pathway flows.

 3)      The introduction provides nice overview of bilirubin and biliverdin generation. However, the heme catabolic pathway also produces carbon monoxide (CO) and iron. Especially since CO signaling mechanisms and iron's involvement in oxidative stress impact many of the subsequent disease associations, acknowledging their production in the introduction is also of merit. Additionally, a simple illustrated overview of the full heme catabolic pathway denoting the structures and biology of heme, biliverdin, bilirubin, CO and iron would help orient readers on how they relate. Establishing all molecules generated and their interconnections early on would provide useful background.

 4)      While the extensive range of natural compounds covered in this review is a strength, the introductions to some of the substances could benefit from a bit more context before detailing particular impacts. For instance, in section 3.1.4, madecassoside is launched right into its effect in an ALS mouse model without first describing its origins or biological nature. Other compounds face this same issue as well. Ensuring every natural modulator gets a brief background sentence or two explaining the basic points of what it is and where it comes from would help readers anchor studies to a foundation of understanding.

 5)      While the content within subsections under “Natural compounds targeting the heme catabolic pathway” highlights useful mechanisms, the presentation is highly condensed, affecting readability. For example, the paragraph on BLVRA modulation begins abruptly by stating “Interestingly, also BLVRA expression is inducible by natural products...” without any lead-in on or why it matters. Other subsections face similar issues, launching straight into details without establishing context. To aid reader comprehension and flow, consider expanding introductions to each mechanism. Add 1-2 sentences for orientation introducing the target, why it is relevant, and how natural modulators interact before detailing specifics.

Comments on the Quality of English Language

NA

Reviewer 2 Report

Comments and Suggestions for Authors

The review by Jayanti et al., well describes the potential effects of several natural compounds on heme catabolic pathway, highlighting the different role of Nrf2, heme oxygenase, bliverdin reductase and UDP-glucuronosyltransferase. Although some natural compounds were mentioned in this review, I suggest to insert additional references for example on mangiferin, ginseng, luteolin, ellagic acid. Cancer is a keyword on this review, therefore i suggest to add more references on cancer and natural products highlighting their apoptotic or ferroptotic effects in vitro and in vivo.

  Comments on the Quality of English Language

Well written

Round 2

Reviewer 1 Report

Comments and Suggestions for Authors

The authors have successfully addressed all the comments.